# The ecological footprint of physicians: A survey of physicians in Canada, India, and USA

**Faramarz Jabbari-Zadeh[1], Arsha Karbassi[2], Aditya Khetan[2]\***

**1** Post-Graduate Medicine, Internal Medicine, Schulich School of Medicine & Dentistry, London, Ontario, Canada, **2** Department of Medicine, Michael G. DeGroote School of Medicine, McMaster University, Hamilton, Ontario, Canada

\* khetana@mcmaster.ca

**Data Availability Statement:** All relevant data are within the paper and its Supporting Information files.

**Funding:** The author(s) received no specific funding for this work.

## Abstract

Combating climate change may be the greatest public health opportunity of the 21st century. While physicians play an important role in addressing climate change, given their affluence in society, they may be an important source of greenhouse gas emissions themselves. We sought to examine the size and nature of the ecological footprint of physicians and medical students. We conducted an online survey from December 2021-May 2022 examining resource consumption, changes in consumption patterns over time, and beliefs about climate change. Participants were medical students, residents, and staff physicians in Canada, India, or USA. Only 20 out of 162 valid respondents had a low ecological footprint (12%), defined as meat intake $\leq$2 times per week, living in an apartment or condominium, and using public transport, bicycle, motorcycle or walking to work. 14 of these 20 participants were from India. 91% of participants were open to reducing their own ecological footprint, though only 40% had made changes in that regard. 49% participants who discussed climate change at work and at home had decreased their ecological footprint, compared to 29% of participants who rarely engaged in such conversations (OR 2.39, 95% CI 1.24–4.63, $P$ = 0.01). We conclude that physicians have a large ecological footprint, especially those from Canada and USA. A majority of physicians are interested in reducing their ecological footprint, and those who engage in conversations around climate change are more likely to have done so. Talking frequently about climate change, at work and at home, will likely increase climate change action amongst physicians.

## Introduction

Combating climate change is likely the greatest public health opportunity of the 21st century [1]. Physicians have historically been staunch advocates for human health on pivotal issues such as nuclear warfare, tobacco, and COVID-19 [2–4]. Physician advocacy on climate change is widespread and an increased focus on the health impact of climate change has likely spurred climate change mitigation [5–8]. However, physicians are usually amongst the wealthiest in most societies (with many in the top 1% globally by income and/or wealth), and medical learners often come from high socioeconomic backgrounds [9–12]. 36–45% of global GHG emissions are contributed by households with incomes in the top 10%, with two thirds of these

**Competing interests:** The authors have declared that no competing interests exist.

households located in high-income countries [13]. Given their affluence in society, physicians may be an important source of GHG emissions themselves.

Ecological footprint refers to how much of the environment is currently needed to support a population's consumption of resources and production of waste [14]. To our knowledge, little research has been done to examine the ecological footprints of physicians and medical students. The purpose of this study was to examine the size and nature of the ecological footprints of physicians and medical students, and to determine their beliefs and attitudes towards climate change mitigation. We believe this is important to investigate as it may lead to greater self-reflection in the physician community about their contribution to ecological change, and their role in societal transition to sustainability. We hypothesized that physicians and medical students would have high ecological footprints, with many taking steps to decrease their individual footprint.

## Methods

### Study design and participants

We conducted a 19-question online survey from December 2021-May 2022 amongst medical students, residents/fellows, and physicians from Canada, India, and USA. We chose to include India to understand potential response differences between high- and middle-income countries. SurveyMonkey was the online software used. A pilot was conducted on 10 participants to ensure clarity of prompts and questions. Survey questions were modified and finalized based on the pilot feedback, before launching the survey. The authors did not have access to information that could identify individual participants during or after data collection.

Participants were recruited via word of mouth (i.e., contacting people in our various social networks who then informed their own networks about the study). Participants were surveyed for resource consumption in food, building, and transport sectors, along with changes in consumption patterns over time. We also surveyed beliefs and attitudes towards climate change mitigation. Demographic information included age, stage of training, and country of residence. All responses were anonymous. Informed consent was gained from all participants and the study protocol was approved by the Hamilton Integrated Research Ethics Board (HiREB). The full set of survey questions can be found in the S1 Appendix.

### Statistical analysis

Based on an estimated population of 2.1 million physicians across Canada, India, and USA, we calculated a sample size of 151 participants. This sample size leads to a ±8% margin of error at a 95% confidence level.

Results are presented as categorical data in percentages. Bar charts were used to visualize proportions. Proportions were compared using Chi-squared tests. Logistic regression models were used to identify participant characteristics independently associated with the odds of reducing one's ecological footprint, including dairy and meat intake. A low ecological footprint was defined as meat intake ≤2 times per week, living in an apartment or condominium and using public transport, bicycle, motorcycle or walking to work. Any participant that did not meet criteria for low ecological footprint was classified as having a high ecological footprint. We used STATA 16.1 software for the analyses.

## Results

There were 179 total responses (response rate of 24.8%) of which 162 were included in our analysis. Participant characteristics are listed in Table 1. Exclusion criteria were not giving consent, not answering questions related to demographics (age, stage of training, and country of

**Table 1. Characteristics of participants.**

| | All respondents (n = 162) | India (n = 49) | Canada (n = 57) | USA (n = 56) |
|---|---|---|---|---|
| **Career Stage** | | | | |
| Medical Student /Resident | 96 (59%) | 16 (33%) | 47 (82%) | 33 (59%) |
| Staff Physician | 66 (41%) | 33 (67%) | 10 (18%) | 23 (41%) |
| **Age Group** | | | | |
| <35 years | 122 (75%) | 32 (65%) | 53 (93%) | 37 (66%) |
| ≥ 35 years | 40 (25%) | 17 (35%) | 4 (7%) | 19 (34%) |
| **Place of Residence** | | | | |
| Apartment/Condominium | 91 (57%) | 32 (65%) | 27 (47%) | 32 (57%) |
| Semi-detached house /rowhouse/townhouse | 18 (11%) | 2 (4%) | 8 (14%) | 10 (18%) |
| Detached house | 51 (32%) | 15 (31%) | 22 (39%) | 14 (25%) |
| **Transportation to work** | | | | |
| Public transport/walking /bicycle | 39 (24%) | 13 (27%) | 12 (21%) | 14 (25%) |
| Motorcycle | 12 (7%) | 12 (25%) | 0 | 0 |
| Car | 111 (69%) | 24 (48%) | 45 (79%) | 42 (75%) |
| **Dietary Style** | | | | |
| No or little eggs/dairy, no meat | 21 (13%) | 11 (23%) | 3 (5%) | 7 (13%) |
| Vegetarian (frequent eggs/dairy, no meat) | 30 (19%) | 18 (37%) | 2 (4%) | 10 (18%) |
| Occasional meat (1–2 portions /week) | 28 (17%) | 10 (21%) | 3 (5%) | 15 (27%) |
| Frequent meat (3–5 portions /week) | 45 (28%) | 7 (15%) | 19 (33%) | 19 (33%) |
| Daily meat intake | 37 (23%) | 2 (4%) | 30 (53%) | 5 (9%) |
| **Locus of responsibility for climate change** | | | | |
| Only corporations and government | 8 (5%) | 1 (2%) | 4 (7%) | 3 (5%) |
| Only individuals | 30 (19%) | 18 (37%) | 1 (2%) | 11 (20%) |
| Change is needed at both levels | 124 (76%) | 30 (61%) | 52 (91%) | 42 (75%) |
| **Open to reducing ecological footprint** | | | | |
| Yes, and have already taken steps | 65 (40%) | 18 (37%) | 22 (39%) | 24 (45%) |
| Yes, but I have not taken significant steps yet | 83 (51%) | 25 (51%) | 31 (53%) | 27 (48%) |
| No, I am not able to in current circumstances | 11 (7%) | 6 (12%) | 2 (4%) | 3 (5%) |
| No, I do not think it will make a difference | 3 (2%) | 0 | 2 (4%) | 1 (2%) |
| **Frequency of climate change mentions in work conversations** | | | | |
| Rare or Never | 59 (36%) | 16 (33%) | 21 (37%) | 22 (39%) |
| 3–12 times per year | 74 (46%) | 21 (43%) | 29 (51%) | 24 (43%) |
| > once a month | 24 (15%) | 10 (20%) | 6 (11%) | 8 (14%) |
| > once a week | 5 (3%) | 2 (4%) | 1 (2%) | 2 (4%) |
| **Frequency of climate change mentions in conversations with family & friends** | | | | |
| Rare or Never | 55 (34%) | 17 (35%) | 14 (25%) | 24 (43%) |
| 3–12 times per year | 67 (42%) | 19 (40%) | 28 (49%) | 20 (35%) |
| > once a month | 24 (15%) | 10 (21%) | 7 (12%) | 7 (13%) |
| > once a week | 15 (9%) | 2 (4%) | 8 (14%) | 5 (9%) |

residence), and not being a practicing physician or a medical trainee. The respondents were primarily young physicians in training, with 122 (75%) of them <35 years of age and 96 (59%) in medical school, residency, or fellowship.

## Buildings

Despite having a higher mean number of people living in their household, a greater proportion of participants from India lived in an apartment or condominium. Fig 1 shows that a higher

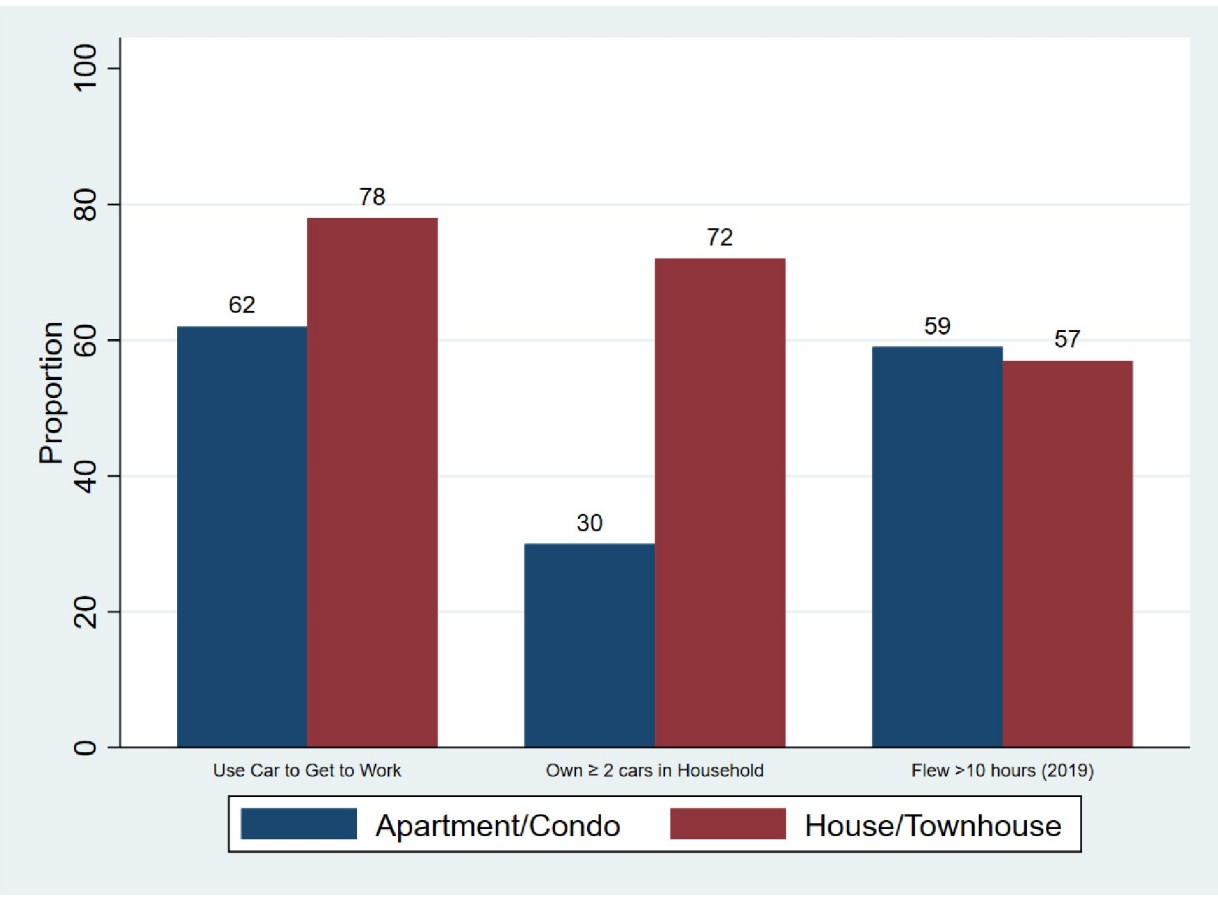

**Fig 1. Increased transport related ecological footprint in house dwellers.**

proportion of participants who lived in a house (detached, semi-detached or townhouse) used a car to get to work and owned ≥2 cars per household, compared to those who lived in an apartment/condominium. A similarly high proportion of participants in both groups flew >10 hours in 2019.

## Transportation

Nearly three quarters of participants from Canada and USA used a car to get to work, with the rest relying on public transport, a bicycle or walking to work. In India, a quarter of participants used a motorcycle to get to work, a common mode of transport in India. Indian participants were also more likely to live in a car-free household, though the absolute frequency was only 10 (20%). On the other hand, 31 (54%) Canadian participants and 30 (54%) American participants had two or more cars in their household, despite having fewer people per household. 6.8% (11/162) of participants owned an electric car. One third of participants from India did not fly at all in 2019, while around two thirds of participants from USA and Canada flew more than 10 hours in the same year.

## Food

Around half the participants ate meat ≤2 times per week, which is consistent with the recommendations of the planetary health diet [15]. However, there were important country-based

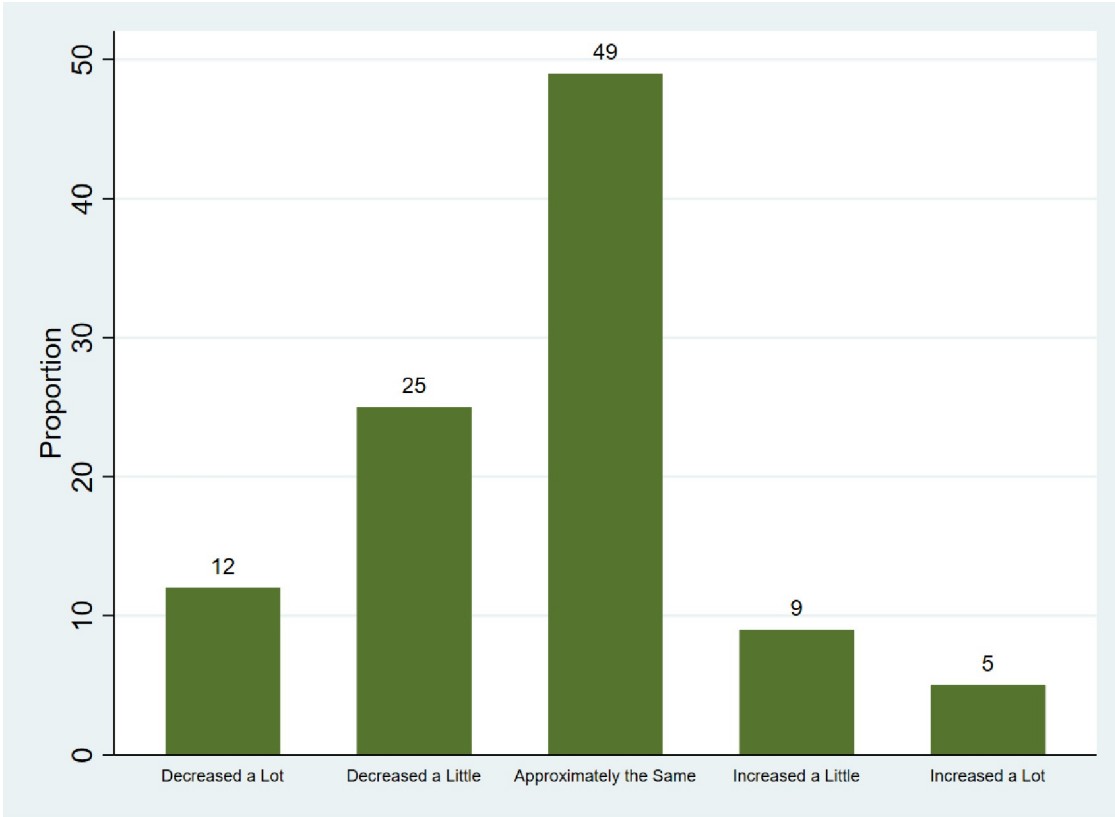

**Fig 2. Change in meat and dairy consumption in last 5 years.**

differences. While 9 (19%) participants from India exceeded this threshold for meat intake, 49 (86%) participants from Canada and 24 (42%) participants from USA exceeded this threshold. Notably, 26 (47%) participants from USA had decreased their consumption of meat and dairy in the past 5 years. Fig 2 shows the change in meat and dairy consumption over five years, for all participants.

## Ecological footprint

Only 20 participants (12%) had a low ecological footprint, which was defined as meat intake ≤2 times per week, living in an apartment or condominium and using public transport, bicycle, motorcycle or walking to work. Compared to participants from Canada and USA, more participants from India had a low ecological footprint $(P < 0.0001)$. 14 of 20 participants with a low ecological footprint were from India.

## Beliefs and attitudes

Three quarters of participants agreed that we need a system change rather than individual change, but we cannot have one without the other, and that individual action plays a critical role in achieving structural change. Consistent with this, 148 (91%) participants were open to reducing their ecological footprint, though a majority had taken few or no steps towards that end. Two thirds of participants rated their ecological footprint as comparable to their peers, while 46 (28%) rated their footprint as smaller than their peers and only 9 (6%) participants believed that their footprint was larger than their peers.

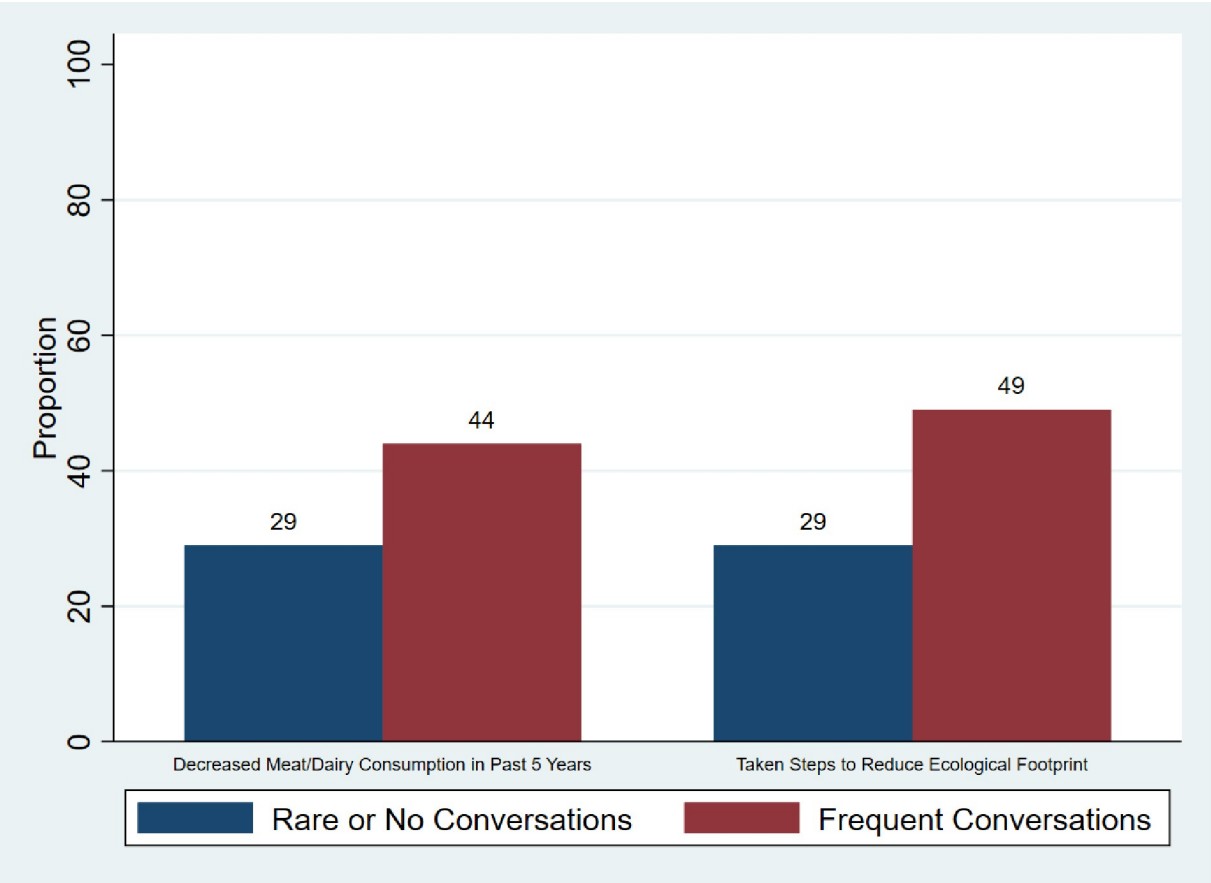

**Fig 3. Association between frequent climate change conversations and changes in ecological footprint.** Participants who engaged in climate change conversations ≤2 per year, either at work or at home, were categorized under 'Rare or No Conversations'. The remaining were categorized under 'Frequent Conversations'.

### Conversations around climate change

Around one third of participants rarely engaged in conversations around climate change, a proportion that was similar for both work and for home. Less than a quarter of participants engaged in climate change conversations more frequently than once a month. Fig 3 shows that a greater proportion of participants who engaged in climate change conversations reduced their ecological footprint, including their meat and dairy intake, compared to participants who rarely talked about climate change.

Participants who engaged in conversations around climate change had significantly higher odds of reducing their ecological footprint (OR 2.39, 95% CI 1.24–4.63, $P = 0.01$), including their intake of meat and dairy (OR 1.92, 95% CI 1.00–3.73, $P = 0.05$).

Supplementary survey results can be found in the Supplementary Table within the S1 Appendix.

### Discussion

Our study has several important findings. First, physicians and medical students have a high ecological footprint, particularly in Canada and USA. Participants from India had a lower ecological footprint than those from Canada or USA as they were more likely to live in an

apartment/condominium, consume meat ≤2 times per week, utilize low carbon modes of transportation (public transport, motorcycle, bicycle, walking), and travelled by air less frequently. This is consistent with population level data from these countries, with average per capita $CO_2$ emissions in USA and Canada nearly 15 tonnes, compared to 2 tonnes in India [16]. Second, a large majority of physicians believed that individual action is important in mitigating climate change, and can help achieve structural change. Consistent with this, >90% of physicians were open to reducing their ecological footprint. Third, physicians who engaged in frequent conversations around climate change were more likely to have taken actions to reduce their ecological footprint, including reducing their intake of meat and dairy.

The buildings sector accounts for 21% of global GHG emissions [17]. Large suburban homes distant from commercial activity and public transportation result in greater GHG emission than apartments near the core of cities. This is largely due to greater energy consumption within large suburban homes and use of cars for daily travel [18, 19]. In our study, participants that lived in houses and townhouses were more likely to own ≥ 2 cars and use a car to travel to work, compared to respondents who lived in apartments. Physicians and medical students, particularly in Canada and USA (where urban sprawl is common), can lower their ecological footprint by living in apartments, instead of detached houses [20]. This may have health benefits for physicians and their families, given that urban sprawl has been linked with obesity, physical inactivity, and decreased life expectancy [21, 22]. For physicians living in detached houses, electrification of heating (using heat pumps) and cooking can be an effective method to decrease their GHG emissions, while improving ambient and indoor air quality [23].

Transportation accounts for 15% of global GHG emissions [24]. Two top contributors of GHG emissions in this sector are road transport and aviation. A majority of participants in our study, especially those in Canada and USA, used cars as their predominant mode of transportation to work. Utilizing low-carbon forms of transportation (such as public transport) and carpooling are two effective alternatives. Given that a large number of car trips are for short distances (<5 km/3mi), using active travel (i.e., walking, bicycling) instead could reduce GHG emissions, while also decreasing the risk of diabetes and cardiovascular disease [25]. Carpooling can be effective for many physicians and medical students, as many of them work together in a central location. Adding one additional passenger per ten vehicles could reduce annual gasoline consumption by 7.5 billion gallons in the United States alone [26]. <10% of study participants owned an electric car. As electric car prices continue to decline, physicians are well positioned, given their relative affluence, to lead the societal switch from internal combustion engines to electric vehicles. However, electric vehicles are not a silver bullet for transport emissions, as they alone will not meet mitigation targets [27]. Reducing vehicle ownership and reducing usage remain key to reducing transport emissions, and also present an opportunity to enhance cardiometabolic health.

A majority of respondents in our study flew more than 10 hours annually prior to the pandemic. This is in sharp contrast to the proportion of the general population that uses air travel, as only 11% of the global population flew in 2018 and the average citizen flies once every 22 months [28, 29]. In the USA, 12% of adults who took more than six flights a year accounted for 68% of all flights taken [28]. From our results, it is evident that a substantial proportion of physicians are frequent fliers. To limit air travel, physicians and medical students can attend events, such as medical conferences and residency interviews, virtually. During the COVID-19 pandemic, many medical conferences were conducted on a virtual platform [30, 31]. In addition to producing lower GHG emissions, virtual conferences are less expensive, more accessible, and demonstrate greater geographical diversity among attendees [32–35]. Physician leaders and academic conference organizers should prioritize virtual conferences and meetings, recognizing their responsibility to minimize environmental harm from such meetings

[36]. Similarly, residency interviews during the pandemic have been conducted virtually across Canada and USA [37, 38]. In comparison to in-person interviews, the virtual format has less GHG emissions and is more affordable for applicants, with one study noting a reduction of 6.3 metric tons of CO2 and savings of at least $2000 per applicant [39]. Virtual interviews are also effective in reducing travel barriers for candidates with disabilities and thus help facilitate the development of a workforce that is more inclusive and representative of the population that it serves [40].

The livestock industry is a widely known driver of climate change and is responsible for 14.5% of global GHG emissions [41]. Approximately half of physicians and medical students in our study ate meat ≤ 2 times per week. However, only 14% of Canadian participants ate meat ≤ 2 times per week, in comparison to 52% of the general Canadian population [42]. Interestingly, 47% of American respondents had reduced their meat intake in the past 5 years, which is comparable to the proportion of the general American population that decreased their meat intake during the COVID-19 pandemic (43%) [43]. Replacing meat partly with plant-based foods may decrease GHG emissions by up to nearly 20% and adopting a vegetarian or vegan diet may reduce emissions by 30–50% [44]. Additionally, a reduction in animal farming and meat intake has other benefits, such as limiting deforestation, promoting biodiversity, conserving water, lowering the risk of cardiovascular disease and cancer, and increasing protein availability by directing proteins used to feed livestock directly to humans [45, 46].

Most physicians live in high-income households, often with an income in the top 10% of their country. By reducing their GHG emissions, physicians can model a low GHG emissions wealthy lifestyle and create new social norms in this regard. Only 10 to 30% of committed individuals are required to create new social norms, and physicians have historically led changes in social norms [47]. Hospitals were one of the first places in society where norms around tobacco use changed, as many physicians quit tobacco once the data around tobacco and health became clear [48]. A majority of physicians and medical students in our study believed that individual action was important in reducing GHG emissions and were open to adopting a more sustainable lifestyle. However, most participants had taken little concrete action to reduce their own ecological footprint. According to the transtheoretical model of behaviour change, this attitude towards climate change mitigation can be classified as in the 'Contemplation' phase [49]. Moreover, it is evident from our findings that having conversations about climate change facilitates transition to the 'Action' phase. Options for initiating more conversations about planetary health in medicine include invited talks and initiatives to reduce the substantial GHG footprint of healthcare [50]. Climate change should also be included in medical school curriculum, and as a part of continuing medical education (CME). Several medical associations, such as the American Medical Association (AMA), the International Federation of Medical Students' Associations (IFMSA), and the Canadian Federation of Medical Students (CFMS) have already called on medical schools to provide more education regarding climate change mitigation [51–53]. However, a 2020 report found that only 15% of medical schools worldwide taught their students about climate change [54]. Commonly cited barriers to implementation of a formal climate change education curriculum include lack of time, funding, staff, and space in the core curriculum. These challenges may partly stem from an underlying assumption that climate change is more of a social or political issue rather than a health issue [55]. Implementing a robust planetary health curriculum will ultimately require both individual and institutional effort, but effective methods include providing financial incentives to medical schools, making climate change education a mandatory criterion for medical school accreditation, inviting climate scientists and experts to be part of medical school faculties, and emphasizing to staff and administrators that climate change is an existential threat to human civilization.

## Limitations and future research directions

The limitations of our study include a relatively small sample size, as a sample size of 162 has a ±8% margin of error at a 95% confidence level. In the future, it would be worthwhile to have a larger sample size and conduct the survey in a greater number of countries to obtain more comprehensive data. Furthermore, while we took several ex-ante measures to decrease sources of common method bias (piloting questionnaire to reduce complexity and ambiguity, using a short questionnaire and providing anonymity), we cannot exclude residual bias with certainty. Bias associated with social media distribution include response bias, nonresponse bias, and order bias. Participants may have also felt a social pressure to respond in a certain way or may have provided inaccurate answers due to poor memory. To minimize response bias, all responses were anonymous, our questions used neutral wording, and we did not promote a specific viewpoint in the survey or in our communications with respondents. Likewise, the people who chose not to respond to our survey may have differed in important ways from those who did participate, which would lead to a biased sample. For instance, three quarters of our respondents were younger than 35 years of age (with 59% in medical school, residency, or fellowship), so our data may not be adequately representative of older physicians. Future research should aim to recruit older physicians, who may not be adequately sampled by research methods that rely solely on online distribution. The order of our questions and answers may have also prompted participants to give certain responses.

## Conclusions

Given their affluence in society, it is important to understand the ecological footprint of physicians. Through an online survey, this study found that physicians and medical students have a large ecological footprint with high emissions from buildings, transport, and food sectors. Compared to physicians in Canada and USA, physicians from India have a lower ecological footprint. A majority of physicians are interested in reducing their ecological footprint, and those who engage in conversations around climate change at work and at home are more likely to have taken action to reduce their ecological footprint. Our findings show that talking frequently about climate change, at work and at home, will likely increase climate change action amongst physicians. Future research should sample a large number of physicians from more countries, and aim to recruit more older physicians. Given their position in society, physicians are well-positioned to drive changes in social norms that can help create a healthy and sustainable world.

## Supporting information

**S1 Appendix.**
(DOCX)

**S1 Data.**
(XLS)

## Author Contributions

**Conceptualization:** Aditya Khetan.

**Data curation:** Faramarz Jabbari-Zadeh, Aditya Khetan.

**Formal analysis:** Aditya Khetan.

**Investigation:** Faramarz Jabbari-Zadeh, Arsha Karbassi.

**Methodology:** Aditya Khetan.

**Project administration:** Aditya Khetan.

**Supervision:** Arsha Karbassi, Aditya Khetan.

**Writing – original draft:** Faramarz Jabbari-Zadeh.

**Writing – review & editing:** Faramarz Jabbari-Zadeh, Arsha Karbassi, Aditya Khetan.

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
