## [Decision Letter · Decision Letter 0]

11 Jul 2023

PONE-D-23-19372The Ecological Footprint of Physicians: A Survey of Physicians in Canada, India, and USAPLOS ONE

Dear Dr. Khetan,

Thank you for submitting your manuscript to PLOS ONE. After careful consideration, we feel that it has merit but does not fully meet PLOS ONE’s publication criteria as it currently stands. Therefore, we invite you to submit a revised version of the manuscript that addresses the points raised during the review process.

We look forward to receiving your revised manuscript.

Kind regards,

Radoslaw Wolniak, full professor

Academic Editor

PLOS ONE

Journal Requirements:

Additional Editor Comments:

APlease adjust the paper accoring to reviewers comments.. 

Reviewers' comments:

Reviewer's Responses to Questions

**Comments to the Author**

1. Is the manuscript technically sound, and do the data support the conclusions?

Reviewer #1: Yes

Reviewer #2: Yes

2. Has the statistical analysis been performed appropriately and rigorously? 

Reviewer #1: Yes

Reviewer #2: I Don't Know

3. Have the authors made all data underlying the findings in their manuscript fully available?

Reviewer #1: Yes

Reviewer #2: Yes

4. Is the manuscript presented in an intelligible fashion and written in standard English?

Reviewer #1: Yes

Reviewer #2: Yes

5. Review Comments to the Author

Reviewer #1: Dear Authors,

Congratulations for your interesting research.

The title and the abstract coincide with the content of the paper. Keywords are well-chosen.

The problem and goal of the paper are clearly and correctly formulated.

The manuscript is clear, relevant for the field and presented in a well-structured manner.

To achieve the goal of the paper, the applied research methods can be considered correct.

The paper contains a comprehensive reference list. The literature studies presented should be considered sufficient both as to the proper selection of sources and their quantity. The cited references are current.

The manuscript’s results are reproducible based on the details given in the methods section.

The figures and the tables are appropriate. They properly show the data. They are easy to interpret and understand.

Reviewer #2: The work deals with an interesting issue about ecological footprints. In my opinion, this is a timely topic. However, I have a few comments to improve the quality of the article.

- Improve the abstract so that it is an abstract without being broken down into chapters.

- Please explain how the obtained results relate to the carbon footprint of the inhabitants of the given countries?

- Please indicate why this occupational group was studied and not another?

- What does high carbon footprint mean? what was the indicator?

- It should be clearly emphasized in the work that 96/162 people are students. Is the result statistically correct, referring to the research sample?

- Please explain how these groups "food", transport, "Climate change talks" relate to the carbon footprint?

- The discussion of the results is correct.

- The Conclusions chapter is really a summary, not a conclusion. They should be added and specific research limitations and further research plans of the authors in this subject should be indicated.

6. PLOS authors have the option to publish the peer review history of their article (what does this mean?). If published, this will include your full peer review and any attached files.

Reviewer #1: No

Reviewer #2: No

---

## [Author Response · Author response to Decision Letter 0]

27 Jul 2023

Reviewer #1: Dear Authors,

Congratulations for your interesting research.

The title and the abstract coincide with the content of the paper. Keywords are well-chosen.

The problem and goal of the paper are clearly and correctly formulated.

The manuscript is clear, relevant for the field and presented in a well-structured manner.

To achieve the goal of the paper, the applied research methods can be considered correct.

The paper contains a comprehensive reference list. The literature studies presented should be considered sufficient both as to the proper selection of sources and their quantity. The cited references are current.

The manuscript’s results are reproducible based on the details given in the methods section.

The figures and the tables are appropriate. They properly show the data. They are easy to interpret and understand.

Response: Thank you for taking the time to provide a thorough review of our manuscript. We deeply appreciate your comments and encouragement. 

Reviewer #2: The work deals with an interesting issue about ecological footprints. In my opinion, this is a timely topic. However, I have a few comments to improve the quality of the article.

- Improve the abstract so that it is an abstract without being broken down into chapters.

Response: Thank you for taking the time to review our manuscript, and provide constructive feedback. We have now re-formatted the abstract and removed chapters. 

- Please explain how the obtained results relate to the carbon footprint of the inhabitants of the given countries?

Response: We have added the following line with a reference in the first paragraph under Discussion, 

‘This is consistent with population level data from these countries, with average per capita CO2 emissions in USA and Canada nearly 15 tonnes, compared to 2 tonnes in India.’

- Please indicate why this occupational group was studied and not another?

Response: The focus on physicians is because of their unique position in society, as they have been staunch advocates for climate change action, but at the same time, may have a high carbon footprint on account of their wealth/income. We elaborated on this in the Introduction, and have added a line in the introduction to clarify this further, 

‘However, physicians are usually amongst the wealthiest in most societies (with many in the top 1% globally by income and/or wealth), and medical learners often come from high socioeconomic backgrounds’ 

- What does high carbon footprint mean? what was the indicator?

Response: A low ecological footprint was defined as meat intake ≤2 times per week, living in an apartment or condominium and using public transport, bicycle, motorcycle or walking to work. Any participant that did not meet criteria for low ecological footprint was classified as having a high ecological footprint. We have mentioned this in Results under ‘Ecological Footprint’, and have now also added this in the Methods section. 

- It should be clearly emphasized in the work that 96/162 people are students. Is the result statistically correct, referring to the research sample?

Response: Thank you for highlighting this important limitation. In the first paragraph under Results, we have mentioned ‘The respondents were primarily young physicians in training, with 122 (75%) of them <35 years of age and 96 (59%) in medical school, residency, or fellowship.’ 

We have also now added this under Limitations, ‘…the people who chose not to respond to our survey may have differed in important ways from those who did participate, which would lead to a biased sample. For instance, three quarters of our respondents were younger than 35 years of age (with 59% in medical school, residency, or fellowship), so our data may not be adequately representative of older physicians.’ 

- Please explain how these groups "food", transport, "Climate change talks" relate to the carbon footprint?

Response: We categorized total ecological footprint by building, transport, and food sectors, in accordance with IPCC convention for sector-wise emissions. The sub-headings of ‘Beliefs and Attitudes’ and ‘Conversations Around Climate Change’ in Results refers to additional survey questions, and are not a direct source of high ecological footprint. 

We have mentioned this under Methods, 

‘Participants were surveyed for resource consumption in food, building, and transport sectors, along with changes in consumption patterns over time. We also surveyed beliefs and attitudes towards climate change mitigation.’

- The discussion of the results is correct.

Response: Thank you!

- The Conclusions chapter is really a summary, not a conclusion. They should be added and specific research limitations and further research plans of the authors in this subject should be indicated.

Response: As suggested, we have modified the Conclusions chapter accordingly. We have also expanded the section on ‘Limitations and Future Research Direction’, adding a line- ‘Future research should aim to recruit older physicians, who may not be adequately sampled by research methods that rely solely on online distribution.’

---

## [Decision Letter · Decision Letter 1]

31 Aug 2023

The Ecological Footprint of Physicians: A Survey of Physicians in Canada, India, and USA

PONE-D-23-19372R1

Dear Dr. Khetan,

We’re pleased to inform you that your manuscript has been judged scientifically suitable for publication and will be formally accepted for publication once it meets all outstanding technical requirements.

Kind regards,

Radoslaw Wolniak, full professor

Academic Editor

PLOS ONE

Additional Editor Comments (optional):

Reviewers' comments:

Reviewer's Responses to Questions

**Comments to the Author**

1. If the authors have adequately addressed your comments raised in a previous round of review and you feel that this manuscript is now acceptable for publication, you may indicate that here to bypass the “Comments to the Author” section, enter your conflict of interest statement in the “Confidential to Editor” section, and submit your "Accept" recommendation.

Reviewer #1: All comments have been addressed

Reviewer #3: All comments have been addressed

2. Is the manuscript technically sound, and do the data support the conclusions?

Reviewer #1: Yes

Reviewer #3: (No Response)

3. Has the statistical analysis been performed appropriately and rigorously? 

Reviewer #1: Yes

Reviewer #3: (No Response)

4. Have the authors made all data underlying the findings in their manuscript fully available?

Reviewer #1: Yes

Reviewer #3: (No Response)

5. Is the manuscript presented in an intelligible fashion and written in standard English?

Reviewer #1: Yes

Reviewer #3: (No Response)

6. Review Comments to the Author

Reviewer #1: In my opinion, the article meets the requirements for writing scientific articles. All comments have been applied. I recommend publication of the article in its present form.

Reviewer #3: (No Response)

7. PLOS authors have the option to publish the peer review history of their article (what does this mean?). If published, this will include your full peer review and any attached files.

Reviewer #1: No

Reviewer #3: No

---

## [Editor Report · Acceptance letter]

4 Sep 2023

PONE-D-23-19372R1 

The Ecological Footprint of Physicians: A Survey of Physicians in Canada, India, and USA 

Dear Dr. Khetan:

I'm pleased to inform you that your manuscript has been deemed suitable for publication in PLOS ONE. Congratulations! Your manuscript is now with our production department. 

Kind regards, 

on behalf of

Professor Radoslaw Wolniak 

Academic Editor

PLOS ONE